# Enhanced YOLOv8-ECCI Algorithm for High-Precision Detection of Purple Spot Disease in Soybeans

**DOI:** 10.3390/s25164958

**Published:** 2025-08-11

**Authors:** Zhihua Deng, Shuyao Ye, Chunru Xiong

**Affiliations:** College of Computer Science and Engineering, Yangjiang Campus, Guangdong Ocean University, Yangjiang 529500, China; 44031106dd@std.gdou.edu.cn (Z.D.); 17875069545@stu.gdou.edu.cn (S.Y.)

**Keywords:** soybean disease detection, YOLOv8, edge detection, deformable convolution, CARAFE, Wise-IoU

## Abstract

**Highlights:**

**What are the main findings?**
Proposed YOLOv8-ECCI achieves 3.0% higher precision and 3.6% better recall for soybean purple spot detection than YOLOv8n.Cross-dataset validation shows 6.0% precision gain and 2.9% mAP@0.5 improvement on African Wildlife data, confirming superior generalization.

**What is the implication of the main finding?**
Enables seed-level disease diagnosis (beyond leaf-level) by solving spectral interference and dense occlusion in stacked soybeans.Provides first dense soybean disease dataset and lightweight model for portable agricultural devices.

**Abstract:**

Seed-level disease detection in soybeans presents significant challenges, including small-sample limitations, spectral interference, and dense occlusions, which are less pronounced in leaf-level analysis. To overcome these obstacles, we propose YOLOv8-ECCI, an enhanced algorithm based on YOLOv8 for high-precision identification of purple spot disease directly on soybean seeds. Experimental results demonstrate that YOLOv8-ECCI substantially outperforms the baseline YOLOv8n model, achieving significant gains of +5.7% precision, +6.5% recall, +8.0% mAP@0.5, and +7.1% mAP@0.5:0.95. Crucially, the model exhibits superior generalization capability, validated through rigorous cross-dataset testing on the African Wildlife dataset, where it surpasses conventional methods by +6.0% precision and +2.9% mAP@0.5. These results confirm that YOLOv8-ECCI effectively addresses the critical challenges in seed-level pathology, providing a robust and accurate solution for practical in-field agricultural disease detection and quality control.

## 1. Introduction

Soybean, a globally vital grain crop, plays a crucial role in food security and agricultural production. Effective disease detection in soybeans is, therefore, of paramount importance. Current research on soybean disease identification predominantly focuses on leaf lesion detection. For instance, the ResNet50V2 model achieved a training accuracy of 97% and a validation accuracy of 96% in classifying soybean leaf diseases [1]. VGG16 and VGG19 algorithms attained accuracies of 99% and 98%, respectively, in soybean disease image classification tasks [2]. Models combining Federated Learning (FL) and Convolutional Neural Networks (CNNs) demonstrate high classification accuracy (averaging around 93%) for soybean leaf diseases, alongside strong generalization capabilities and robustness [3,4]. Some studies have employed methods like the Gray-Level Co-occurrence Matrix (GLCM) to extract image texture features, coupled with optimization algorithms such as the Whale Optimization Algorithm (WOA) for feature enhancement and classification, thereby improving model accuracy, sensitivity, and specificity [5]. Research has also addressed analyzing and grading disease severity to provide more targeted solutions for precision agriculture and disease management. Furthermore, domestic research in China has made significant strides in soybean seed recognition and optimizing neural networks using intelligent algorithms. Studies based on YOLO algorithms, such as utilizing YOLOv8 for soybean disease detection and employing intelligent optimization algorithms to enhance complex CNNs, have achieved high-precision identification of soybean leaf diseases [6].

However, research specifically targeting disease detection on soybean seeds themselves remains notably insufficient, with the limited relevant literature.

Soybean purple stain disease, caused by the fungus Cercospora kikuchii, is a globally significant fungal disease posing a substantial threat to soybean seed quality, market value, and agricultural production [7]. First identified on the Korean Peninsula in 1921, the disease has now spread to major soybean-producing regions worldwide. The pathogen can infect soybean leaves, stems, pods, and seeds. Purple stain symptoms on seeds not only directly downgrade their market grade but also impair seed physiological activity and processing quality. With the expansion of global soybean trade and increasing disease dissemination, the development of early detection technologies targeting seed health is crucial for safeguarding food security and agricultural product quality.

Detecting diseases in stacked soybean samples within laboratory environments presents particular core challenges:Small-Sample Overfitting Risk: Annotated data for seed diseases is scarce. Traditional models (e.g., Faster R-CNN), reliant on large datasets, are prone to overfitting with limited samples.Complex Spectral Interference: Seed stacking causes uneven light reflection and color mixing between lesions and healthy areas (e.g., mold-induced yellow spots vs. natural yellow seed coats), adversely affecting the robustness of traditional spectral analysis methods.Missed Detection due to Dense Occlusion: In stacked seed scenarios, disease regions experience high occlusion rates. Existing models exhibit limited capability in capturing subtle local features, with reported missed detection rates exceeding 30% in dense scenes.

This study proposes an improved YOLOv8 model to address these key technical challenges in soybean seed disease detection, aiming to advance the field from “leaf-level” to “seed-level” identification. YOLOv8, as a significant iteration of the YOLO series, offers several advantages pertinent to this task:Small-Sample Learning: Leveraging adaptive data augmentation (Mosaic, MixUp) and transfer learning (ImageNet pre-trained weights), YOLOv8 enhances model generalization and mitigates overfitting with limited annotated data.Complex Feature Resolution: Its enhanced backbone network supports multi-scale feature fusion. Combined with dynamic convolution modules, it effectively distinguishes spectral ambiguities between lesions and healthy tissue (e.g., differing texture roughness in moldy regions).Dense Occlusion Optimization: The integration of attention mechanisms strengthens the weighting of local lesion features. Loss function modifications further optimize the localization accuracy of occluded targets, reducing missed detection rates.Application Suitability: YOLOv8’s lightweight design and high inference speed align with the computational constraints of mobile devices, laying the foundation for developing portable laboratory detection equipment.

## 2. Related Work

In this section, we briefly introduce some work relevant to this study. Section 2.1 will review work related to soybean data sources. Next, Section 2.2 will cover techniques associated with data preprocessing. Finally, Section 2.3 will provide a brief discussion of the YOLO framework.

### 2.1. Data Sources

Soybean disease image data were collected at the Yangjiang Customs, China, on 3 April 2024, between 15:00 and 17:00 (UTC + 8). All images were captured in a laboratory setting with a fixed camera position at a resolution of 3024 × 4032 pixels. To simulate real-world conditions and enhance data diversity as well as model generalization capability, the soybean samples were systematically repositioned for each capture.

The primary symptom of the soybean disease, identified and confirmed by plant pathology experts, is purple blotching. An initial dataset of 164 valid images was obtained. Part of the dataset is shown in Figure 1.

### 2.2. Data Preprocessing

In this study, offline data augmentation techniques were employed to expand and enrich the training dataset. Initially, the sample images within the dataset were precisely annotated using the LabelImg (v1.8.1) annotation software, constructing a sample set with labeled disease regions. To mitigate the risks of overfitting and poor generalization capability arising from insufficient dataset size, offline data augmentation was applied to the image dataset prior to training. Subsequently, a series of augmentation operations were implemented, including brightening, shearing, affine transformations, random flipping, HSV contrast adjustment, and translation, to enhance the geometric and color diversity of the images. Representative examples of augmented image data are shown in Figure 2.

Finally, image normalization was performed by adjusting pixel values using mean and standard deviation parameters, promoting a more uniform distribution to facilitate training stability and accelerate convergence. Through these offline data augmentation techniques, the training dataset was effectively expanded, significantly improving the model’s generalization capability and robustness, thereby establishing a solid foundation for subsequent model training and performance evaluation.

### 2.3. YOLOv8 Framework

YOLOv8, a significant iteration within the YOLO series, incorporates improvements and optimizations in network architecture, loss functions, training strategies, and performance, demonstrating distinct advantages [8].

Regarding network architecture, YOLOv8 introduces the C2f module, inheriting the CSP concept. This module reduces both parameter count and computational complexity, achieving a lightweight design. This enhances applicability on resource-constrained devices while maintaining high accuracy. Furthermore, YOLOv8 retains the PAN (Path Aggregation Network) concept for feature fusion but optimizes efficiency by eliminating redundant convolutional structures in the upsampling stage. It replaces the C3 module with the C2f module, thereby enhancing feature fusion efficiency and strengthening the model’s ability to represent features of objects across different scales, ultimately improving detection performance.

Concerning loss functions and training strategies, YOLOv8 employs a combination of Varifocal Loss (VFL) for classification and Distribution Focal Loss (DFL) alongside Complete Intersection over Union (CIoU) Loss for regression. This combination optimizes the training process, improves the model’s learning of target features, and enhances detection accuracy and robustness. Simultaneously, YOLOv8 incorporates an adaptive anchor frame technique. This technique automatically optimizes preset bounding box parameters, boosting the model’s adaptability across diverse scenarios, reducing the need for manual intervention, and improving usability.

## 3. Method

The structure of this section is as follows: Section 3.1 provides an overview of the methodology employed in this study, detailing how the individual modules are integrated to form the novel architecture of the improved model. Subsequently, Section 3.2 presents the overall design of the EIEStem module, Section 3.3 describes the overall design of the C2f_DCNv3 module, and Section 3.4 outlines the overall design of the CARAFE module. Finally, Section 3.5 details the overall design of the Wise-IoU loss function.

### 3.1. Overview

To address the prevalent challenges in soybean disease detection—namely model overfitting, missed detections, and low detection accuracy—this study proposes a high-precision soybean disease detection method based on an enhanced YOLOv8 framework. The proposed approach incorporates four key optimizations to achieve superior detection performance with reduced computational costs:Mitigating Overfitting: The C2f_DCVv3 module, integrating DCNv3 convolution (Deformable Convolution v3), is introduced to effectively handle deformed targets within images and extract features with greater precision [9]. Concurrently, the structural design of the Bottleneck module helps mitigate gradient explosion issues while fully preserving feature information, thereby enhancing the model’s recognition performance and efficiency for soybeans infected with purple spot disease.Reducing Missed Detections: The EIEStem module is incorporated to capture richer information from images. Its core component, the Sobel operator, exhibits inherent noise suppression capabilities, effectively reducing interference caused by uneven illumination or surface texture noise [10].Improving Detection Accuracy: To aggregate contextual information over a large receptive field, the CARAFE (Content-Aware ReAssembly of FEatures) module is leveraged. Its large receptive field characteristic allows aggregation of more contextual information, enabling better capture of subtle disease features on soybeans and improving recognition accuracy [11]. CARAFE’s lightweight design ensures model efficiency, facilitating the processing of large-scale soybean image datasets. Its content-aware processing capability dynamically generates upsampling kernels based on specific soybean features, further boosting recognition accuracy.Enhancing Bounding Box Regression: The Wise-IoU (WIoU) loss function is adopted to improve the network’s bounding box regression performance. WIoU assists the model in more accurately identifying soybeans with purple spot disease, even in dense clusters with similar coloration, thereby increasing detection accuracy and model robustness [12].

The final optimized model is termed the YOLOv8_ECCI network. Its architecture is illustrated in Figure 3.

### 3.2. EIEStem Module

The EIEStem module integrates multiple feature extraction techniques to capture rich information in images, with the Sobel operator as its core component. First, the Sobel operator involves only simple convolution operations, resulting in low computational complexity and high execution efficiency [13]. In particle-level soybean disease recognition, large volumes of image data must be processed. The Sobel operator’s efficiency helps reduce computation time, thereby enhancing real-time performance and practical applicability. In soybean grain images, the grayscale changes between diseased and healthy regions may be relatively gradual. It can effectively detect such gradual edges, showing strong robustness in grayscale gradient images and improving disease region identification accuracy [14]. The detection results are shown in Figure 4.

Secondly, the Sobel operator computes edge gradients by calculating weighted differences in grayscale values of adjacent pixels. This design provides inherent noise-smoothing capability, effectively reducing its impact on edge detection [15]. In soybean disease identification, soybean grain images may be affected by noise such as lighting and dust. The Sobel operator can better extract real edge information, thereby improving the accuracy of disease feature recognition.

Finally, the Sobel operator detects horizontal and vertical gradient changes and computes pixel-level gradient directions. By enhancing edge features through gradient calculations, it makes the contours of lesions more distinct [16]. In soybean disease recognition, lesion shape and edge orientation are key features. The Sobel operator provides more directional information for subsequent feature extraction and classification [17]. The principle is shown in Table 1.

$G_{x}=\left[\begin{array}{ccc} -1 & 0 & 1\\ -2 & 0 & 2\\ -1 & 0 & 1 \end{array}\right]\times I$, $G_{y}=\left[\begin{array}{ccc} -1 & -2 & -1\\ 0 & 0 & 0\\ 1 & 2 & 1 \end{array}\right]\times I.$ The gray scale size formula for this point is $G=\sqrt{{G_{x}}^{2}+{G_{y}}^{2}}$. The EIEStem module builds a more comprehensive feature representation by concatenating the output of the SobelConv layer with that of another sequence comprising a zero-padding layer and a max-pooling layer in the channel dimension. This fusion allows the model to capture both rich edge and spatial information, leading to more accurate differentiation between diseased and healthy beans.

Compared to the traditional two Conv modules, the EIEStem module significantly reduces the number of parameters that need to be trained by using shared convolutional layers, thereby reducing the model’s storage and computational overhead and improving computational efficiency. This is particularly important for processing large amounts of image data, especially when performing real-time recognition on resource-constrained devices.

### 3.3. C2f_DCNv3 Module

C2f-DCNv3 is an improved network architecture that combines the C2f (Concatenation with Fusion) module and Deformable Convolutional Networks v3 (DCNv3). It combines feature fusion from C2f with dynamic sampling from DCNv3 to improve the model’s adaptability to complex shapes and scale variations [18]. DCNv3 adjusts the convolution operation through dynamic offsets, enabling the model to better handle changes in target shape and scale. In traditional convolutional networks, the sampling positions of convolutional kernels are fixed, making it difficult to adapt to geometric deformations of targets. DCN introduces learnable offsets, allowing dynamic adjustment of kernel sampling positions to accommodate target deformation [19]. The DCN module consists of an offset prediction network and deformable convolutional operations, as shown in Figure 5.

The offset prediction network uses 3 × 3 convolution layers and ReLU activation functions. Its input is the original input features, and it generates outputs of the same size as the input features. The output represents the offset amount for each input position. Then, based on the predicted offset amounts, a deformable convolution operation is performed on the input features. The deformable convolution operation first calculates the sampling points for each input position, then computes the output features based on the sampling points and offset values. Compared to DCNv1 and DCNv2, DCNv3 demonstrates superior performance and robustness. Its principle is as follows: $y(r_{0})=\sum_{y=1}^{Y}\sum_{z=1}^{Z}k_{y}q_{yz}x_{y}(r_{0}+r_{z}+∆r_{yz})$, where Y denotes the total number of aggregation groups. For group y, k_y_ ∈ R^C×C'^ denotes the position-independent projection weight of the group, where C′ = C/G denotes the group dimension. q_yz_ ∈ R denotes the modulation scalar of the zth sampling point in group y. x_y_ denotes the input feature map of the slices. Δr_yz_ is the offset corresponding to the mesh sampling position r_z_ in group z, and y(r_0_) is the output.

Unlike attention-based operators such as MHSA or deformable attention, the DCNv3 operator preserves convolutional inductive bias, making it more efficient with less training data and faster convergence [20]. The deformation-based convolution (DConv) process is illustrated in Figure 6.

In YOLOv8’s feature extraction network, initial convolutional layers reduce image resolution. However, due to the small size of the infected soybeans, early detection becomes difficult. Additionally, healthy soybeans and those infected with purple spot disease share similar feature information in their main bodies. Therefore, extracting features of soybeans infected with purple spot disease through these two convolutional operations is challenging. Furthermore, the original C2f module contains a large number of parameters. To address these issues, this paper introduces the C2f-DCNv3 module. Each C2f-DCNv3 module consists of two DCNv3 modules and several Bottleneck-j (j = 1, 2) modules. The DCNv3 convolution effectively handles deformable objects in images, aiding in the precise extraction of desired features. The Bottleneck-j modules are divided into Bottleneck-I and Bottleneck-2 based on their structure. In Bottleneck-I, the main network is optimized through residual connections to mitigate the gradient explosion problem. Bottleneck-2 uses sequential connections for information flow, enabling more effective retention of feature information from the neck stage.

### 3.4. CARAFE Module

In soybean disease identification, the CARAFE module offers an efficient and accurate upsampling method due to its unique design advantages [21]. The design of the CARAFE module fully considers three key characteristics: large receptive field, content-aware processing, and lightweight computation [22]. The working principle of CARAFE is shown in Figure 7.

The upsampling kernel of CARAFE is not fixed but rather “adaptive”: first, the input feature map is compressed to 64–128 dimensions via a 1 × 1 convolution, followed by a 5 × 5 group convolution to predict a set of 25-dimensional weights at each spatial location. After Softmax normalization, this becomes a “small convolution kernel” whose weights vary dynamically based on the content. Weights sharply increase at lesion edges and gradually even out in smooth leaf regions. During reconstruction, simply apply these dynamic weights to the 5 × 5 neighborhood pixels. This sharpens and preserves lesion boundaries while maintaining smoothness in uniform regions, all achieved with just two lightweight convolutions, decoupling computational complexity from the number of input channels.

First, CARAFE differs from traditional upscaling methods, such as bilinear interpolation, as it can aggregate contextual information within a large receptive domain [23]. This enables CARAFE to better capture fine details and surrounding context, which is critical for identifying densely infected soybeans. Since these targets often occupy small regions, more contextual cues are needed for accurate detection. This wide field of view enables the model to capture broader environmental information, thereby more accurately identifying infected areas.

Second, the CARAFE module’s lightweight design ensures low computational overhead and avoids introducing excessive parameters. This allows it to be easily integrated into existing network architectures while maintaining high efficiency [24]. This is particularly beneficial for processing large-scale soybean datasets, allowing the model to maintain speed and accuracy.

Finally, the content-aware processing capability of the CARAFE module enables the upsampling kernel to be dynamically generated based on the semantic information of the input feature map, rather than using a fixed kernel. This content-based upsampling strategy can better adapt to the specific characteristics of soybeans and improve recognition accuracy. The overall structure of the CARAFE upsampling network is shown in Figure 8.

CARAFE and nn.Upsample (Nearest Neighbor Upsampling) are two distinct upsampling methods used in YOLOv8. The latter preserves edge details better due to its simple replication mechanism. In this study, the CARAFE module is further optimized by integrating the Sobel operator to enhance edge detection, making the structure more efficient and semantically accurate.

### 3.5. Wise-IoU Functions

The Wise-IoU loss is an improved function for bounding box regression in object detection. It addresses the negative impact of low-quality samples through a dynamic non-monotonic focusing factor (FM) [25]. Unlike traditional IoU-based approaches, it adjusts the loss weight based on the degree of deviation (outlier level) of anchor boxes rather than the IoU value itself [26]. Outliers are defined as follow: $\beta =\frac{L_{IoU}}{\bar{L_{IoU}}}$, $r=\frac{\beta }{\delta \alpha^{\beta -\delta }}$, where δ is a constant and β is the ratio of the exponential moving average of the IoU losses and the anchor and target frames $L_{IoU}$ and $L_{IoU}$ so that that a nonmonotonic focusing coefficient r = 1 is constructed when β = δ. The coefficient r is used in the WIoU v1 to obtain the WIoU v3: $L_{WIoU\;v3}=r\cdot L_{WIoU\;v1}$. Wise-IoU evaluates the quality of anchor frames through a dynamic non-monotonic focusing mechanism that uses “outliers” as an alternative to IoU and provides a gradient gain assignment strategy. This strategy reduces the dominance of high-quality anchors, suppresses the harmful gradient influence of low-quality samples, and redirects attention toward medium-quality anchors, thereby improving the detector’s overall performance [27]. Experimental results demonstrate that Wise-IoU has a positive impact in improving the accuracy of purple spot disease soybean recognition. The effect of different loss functions is shown in Figure 9.

## 4. Tests and Analysis

### 4.1. Test Environment and Parameter Configuration

Experiments were conducted on a system equipped with an NVIDIA GeForce RTX 3090 GPU (24 GB VRAM) and Ubuntu 20.04 OS. The model was implemented in Python 3.8 using PyTorch 1.11.0 and trained with CUDA 11.3. The YOLOv8n implementation used the Ultralytics library (v8.2.50). The training parameter settings are shown in Table 2.

The epochs = 300 is required for conditions with weaker features, such as small targets, occlusions, and similar background interference, which necessitate more time to learn subtle features. Sufficient iteration is needed to optimize feature extraction capabilities, and early stopping with patience = 100 is employed to prevent overfitting. The imgsz = 640 is a balanced choice for YOLOv8, balancing detail resolution and computational efficiency, capable of capturing small lesions and sensitive-to-occluded scenes. The batch = 32 + workers = 4 is a common batch size, with large batches reducing gradient noise in occluded samples. The optimizer = SGD + momentum = 0.937 is a classic combination in the YOLO series, widely validated as effective, and more robust against non-smooth optimization issues caused by occlusions. SGD is the standard for object detection, combined with momentum to escape local minima. The lr0 = 0.01 + lrf = 0.01 with slow decay facilitates fine-tuning complex features in later stages. When used with warmup_momentum = 0.8, it effectively avoids the instability caused by high initial LR and prevents early gradient direction errors caused by complex background interference. The weight_decay = 0.0005 is crucial for scenes with high feature interference, preventing overfitting to background noise.

### 4.2. Evaluation Indicators

In this study, precision P, recall R, mean average precision mAP, and the number of parameters were used as evaluation metrics for the model. The definitions of P, R, and mAP are as follows:

$P=\frac{T_{p}}{T_{p}+F_{p}}\times 100\%$, $R=\frac{T_{p}}{T_{p}+F_{N}}\times 100\%$, $AP=\int_{0}^{1}P(r)dr$, $mAP=\frac{1}{Q}\sum_{i=1}^{Q}{AP}_{i}$, where $T_{p}$ is the sample of correct predictions, $F_{p}$ is the sample of predictions that are not for the predicted target as the target,$\;F_{N}$ is the category of predicting the target as something else, AP is the accuracy of individual categories, and mAP is the mean average precision.

The number of parameters refers to the total trainable weights and biases in the model. It reflects the model’s complexity and memory footprint. A lower parameter count is considered a key indicator of model efficiency and compactness.

### 4.3. Test Results and Analysis

#### 4.3.1. Comparative Experimental Analysis of Different Algorithms

To evaluate the performance of the YOLOv8-ECCI model, comparative experiments were conducted under identical conditions using several mainstream models: YOLOv5s, YOLOv7, YOLOv8n, YOLOv7-tiny, and YOLOv8s. Given that the soybean image dataset used in this study is relatively limited in size, and in order to maximize the use of limited samples, obtain stable and reliable model performance evaluations, and reduce the risk of overfitting, this study adopted a k-fold cross-validation (k = 5) strategy for model training and evaluation. The results are presented in Table 3.

YOLOv5s: As a baseline lightweight architecture (depth/width multiplier = 1), its limited receptive field design (Parameters = 4.5 MB) leads to critical shortcomings in densely stacked soybean scenarios:Insufficient deep texture feature extraction capability (Recall = 0.518).High false detection rates in lesion edge blur regions (Precision = 0.647).Failure in multi-scale lesion fusion (mAP@0.5~0.95 = 0.186).

This demonstrates the inherent inability of fundamental architectures to address fine-grained detection requirements in complex agricultural environments.

YOLOv9: While attempting to enhance feature representation through increased parameters (12.7 MB, ↑182%), its redundant computational structure (GELPOs = 22.1) results in the following:Low-efficiency information propagation in feature pyramids (mAP@0.5 = 0.536, ↓17%).Aggravated gradient vanishing for small lesions (Recall = 0.510, ↓11.3%).Imbalance between computational density and accuracy gains (mAP/Parameter = 0.042, ↓38%).

This reflects that merely increasing network depth fails to resolve domain-specific challenges in agricultural applications.

YOLOv8n: In pursuit of deployment efficiency (GELPOs = 6.8), excessive compression of feature channels (Parameters = 5.4 MB) causes the following:Inadequate sensitivity to chromatic differences in lesions (Precision = 0.629, ↓12.4%).Boundary feature confusion in adherent soybeans (mAP@0.5 = 0.540, ↓13.9%).Loss of pathological texture information in shallow networks (Recall = 0.528, ↓8.3%).

This reveals the fundamental trade-off between model lightweighting and feature fidelity [28].

YOLOv10: The dynamic parameter mechanism (Parameters = 5.5 MB) exhibits critical limitations in agricultural contexts:Imbalanced weight allocation due to non-uniform lesion size distribution (mAP@0.5 = 0.522, ↓16.8%).Weak chromatic feature response under low-light conditions (Precision = 0.618, ↓14.0%).Failure of feature reorganization modules (mAP@0.5~0.95 = 0.188, ↓29.6%).

This indicates generic dynamic strategies are ill-suited for agricultural settings with specialized lighting and target distributions.

YOLOv11: Representing the state-of-the-art in lightweight models (Parameters = 5.2 MB, GELPOs = 6.3), it still suffers from the following:Inadequate fusion of deep pathological features (mAP@0.5 = 0.560, ↓10.7%).High miss rates for minute lesions (Recall = 0.527, ↓8.5%).Weak high-precision localization capability (mAP@0.5~0.95 = 0.198, ↓25.8%).

This marks the performance ceiling of conventional lightweight architectures for agricultural disease detection.

Based on earlier experiments, YOLOv8n was selected as the baseline model for detailed comparison. The comparative results are visualized in Figure 10.

In real-world detection scenarios, inconsistency in soybean stack height often leads to variable lighting conditions and changes in surface reflectance, a phenomenon known as spectral variability. Spectral variability is a critical factor in image processing, as it directly affects feature extraction quality and target recognition accuracy. Comparative analysis shows that the YOLOv8-ECCI model exhibits greater robustness to spectral variability, significantly reducing missed and false detections. It also achieves higher average recognition accuracy compared with the YOLOv8n model. This improvement can be attributed to several targeted architectural enhancements. First, YOLOv8-ECCI incorporates the Sobel edge detector and deformable convolution (DCN). The Sobel operator enhances boundary clarity, while the DCN module adapts to geometric variations in target shape. These improvements make the model more accurate and efficient in feature extraction and disease identification. Additionally, it incorporates a high-performance upsampling algorithm and the Wise-IoU loss function. These improvements enhance the model’s adaptability to varying scales and densities, enabling higher precision and recall in detecting purple spot disease under densely packed conditions. As a result, YOLOv8-ECCI effectively captures the features of purple spot disease and maintains high recognition accuracy even in densely populated scenes, demonstrating the innovation and practical value of the proposed improvements in the field of object detection. The k-fold cross-validation strategy was crucial for overcoming data volume limitations, obtaining robust performance evaluations and model selection, and ensuring the reliability of the research conclusions. Future work can further improve model performance by expanding the dataset.

#### 4.3.2. Ablation Test

To evaluate the effectiveness of each proposed improvement, YOLOv8n was selected as the baseline model. The EIEStem module was integrated into the backbone to enhance edge feature extraction. The CARAFE upsampling module was introduced in the neck, and the standard C2f module was replaced with the C2f-DCNv3 module. These modifications aim to enrich feature representations for detecting purple spot disease in soybeans while maintaining model efficiency. Ablation experiments were conducted to test the baseline model and various combinations of the proposed modules individually and jointly. This allowed for assessing the individual contribution of each component to overall model performance. Results are summarized in Table 4.

Table 4 shows that integrating the EIEStem module into the backbone consistently improves performance metrics. This improvement is attributed to the Sobel operator’s ability to enhance edge detection while suppressing noise interference, thereby enabling more accurate identification of purpura in complex environmental conditions.

The application of the C2f-DCNv3 module exhibits certain random fluctuations. Through multiple experiments, we found that deformable convolutions inherently possess randomness, which stems from sampling feature maps via learned dynamic offsets, leading to unstable oscillations in local optima. This results in convergence to different performance points when identifying occluded soybeans in densely arranged scenarios. However, the C2f-DCNv3 module enables the model to better handle changes in target shape and scale, thereby better matching the complexity and diversity of the dataset in this study. In subsequent work, we will introduce offset regularization and progressive fine-tuning strategies to control its random fluctuations and further optimize the model.

Additionally, the CARAFE module, by capturing richer contextual and spatial information, has already positively impacted model performance when applied independently. Experimental results further demonstrate that when CARAFE is combined with the EIEStem and C2f-DCNv3 modules and paired with the Wise-IoU loss function, its advantages become even more pronounced. The synergistic interaction between modules indicates that modular improvements are complementary and can collectively enhance the model’s accuracy and robustness.

#### 4.3.3. Comparative Experimental Validation of Model Generalizability

In order to verify the generalizability of the proposed method in this study, YOLOv8-ECCI and YOLOv8n are compared on other datasets. The open-source global dataset African Wildlife was chosen for this experiment. There are also cases of small target detection for animals in the image, and there are also cases of occlusion for animals densely distributed in the image, which is similar to soybean detection. The test results are shown in Figure 11. Compared with YOLOv8n, YOLOv8-ECCI improves the accuracy by 6 percentage points, which effectively reduces the problem of false and missed detection of animals, and improves the accuracy by 2.9 and 2.1 percentage points on mAP@0.5 and mAP@0.5:0.95, respectively. The results are shown in Figure 11, which indicates that YOLOv8-ECCI has a higher detection accuracy.

The detection results of the two models on the wildlife dataset are shown in Figure 12.

It can be seen that YOLOv8-ECCI also has better detection results on the complex wildlife dataset, and YOLOv8-ECCI can more accurately detect the targets and numbers of buffaloes and elephants, which proves the generalizability of the method of this study.

## 5. Conclusions

In this paper, an improved YOLOv8-ECCI model is proposed for the accurate detection of soybeans affected by purple spot disease. Building on YOLOv8n, the traditional dual convolutional modules are replaced with EIEStem modules. These construct a more complex feature representation by concatenating the outputs of the SobelConv layer and a parallel sequence comprising a zero-padding layer and a max-pooling layer along the channel dimension, thus incorporating rich edge and spatial information for improved recognition of both diseased and healthy soybeans. Deformable convolution (DCN) is introduced to effectively handle deformed targets, improving the model’s adaptability to variations in shape and scale. The CARAFE upsampling operator is employed to improve detection accuracy while reducing model parameters and floating-point operations. The Wise-IoU loss function is introduced to mitigate gradient vanishing and enhance the bounding box regression performance, thereby further improving detection accuracy in complex scenes. Experimental results demonstrate that the proposed YOLOv8-ECCI model achieves high precision and recall, along with superior generalization ability, particularly under conditions involving small sample sizes, spectral variability, and occlusion, aligning well with expectations. The following conclusions can be drawn from the comparative analysis of the experimental results:Under the same test conditions, the improved YOLOv8-ECCI outperforms YOLOv5s, YOLOv9, YOLOv10, YOLOv11, and YOLOv8n on the soybean disease dataset. Compared with the original YOLOv8n baseline model, it achieves improvements of 8.9, 8.7, and 7.7 percentage points in accuracy, mAP@0.5, and mAP@0.5:0.95, respectively. Additionally, the number of parameters is increased by 1.2%, making YOLOv8-ECCI the most compact and accurate model among the compared methods. These results provide methodological support for the rapid detection of soybean diseases.To validate the detection effectiveness of the improved YOLOv8-ECCI network model, this study performs a visual comparative analysis by using publicly available datasets. The results show that the improved YOLOv8-ECCI detection effect is always better than the original YOLOv8n, with better performance in the face of occlusion, spectral diversity, and dense small item recognition, which provides a reference for recognizing more complex disease scenarios in the follow-up.

The main contributions of this study include the following:The construction of the first disease detection dataset for densely stacked soybean grains, which fills a research gap in the field.The development of an edge-enhanced YOLOv8-based model that integrates deformable convolution, a wide receptive field upsampling operator, and a high-performance regression mechanism, offering a practical solution for small-sample agricultural detection tasks.

Currently, the proposed model achieves satisfactory results for the recognition of purple spot disease in soybeans. However, many other types of soybean diseases exist, and issues such as interference from foreign objects and model overfitting caused by multi-class variability have not yet been addressed. The next step will be to enrich the soybean disease samples and supplement the information of the samples that will be detected with foreign objects, so as to do further research on identification and real-time detection.

## Figures and Tables

**Figure 1 sensors-25-04958-f001:**
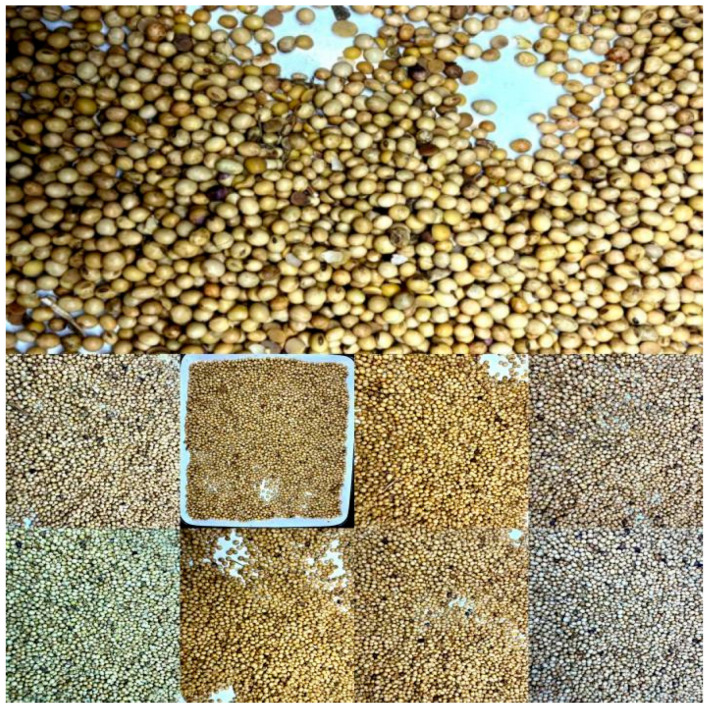
Sample data from soybean disease images. This dataset was subsequently augmented to a total of 900 images using data augmentation techniques. The complete dataset was then randomly partitioned into training, validation, and test sets at a ratio of 7:2:1. The training set was utilized for model training, the validation set for preliminary model performance evaluation and hyperparameter tuning, and the test set for assessing the final model’s classification accuracy and generalization capability.

**Figure 2 sensors-25-04958-f002:**
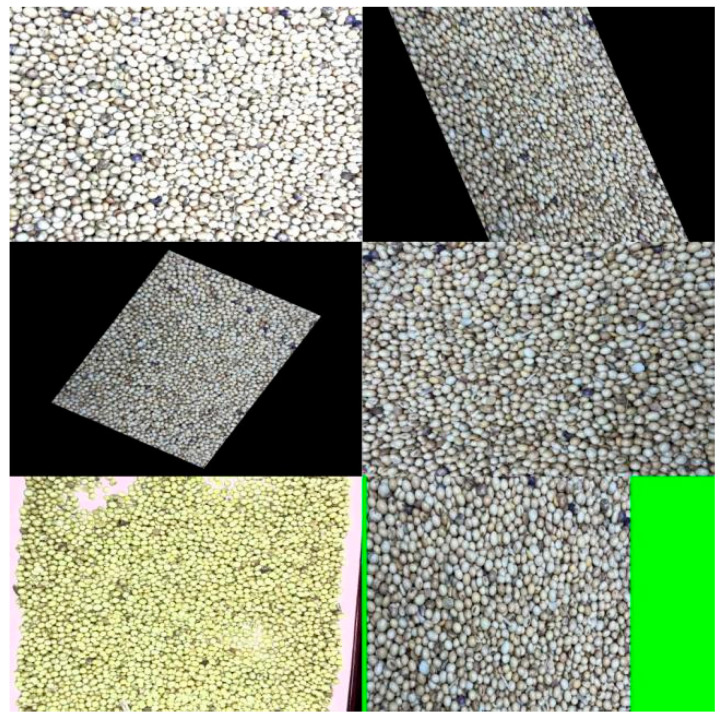
Enhanced Image Sample Data.

**Figure 3 sensors-25-04958-f003:**
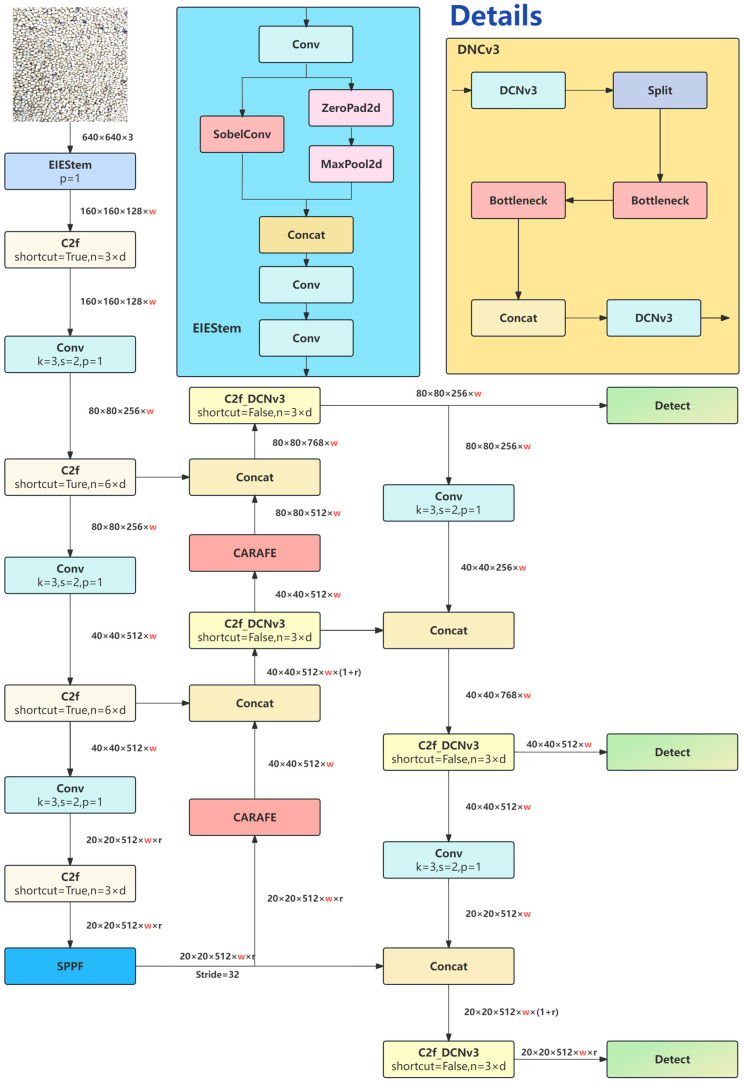
YOLOv8-ECCI network architecture diagram.

**Figure 4 sensors-25-04958-f004:**
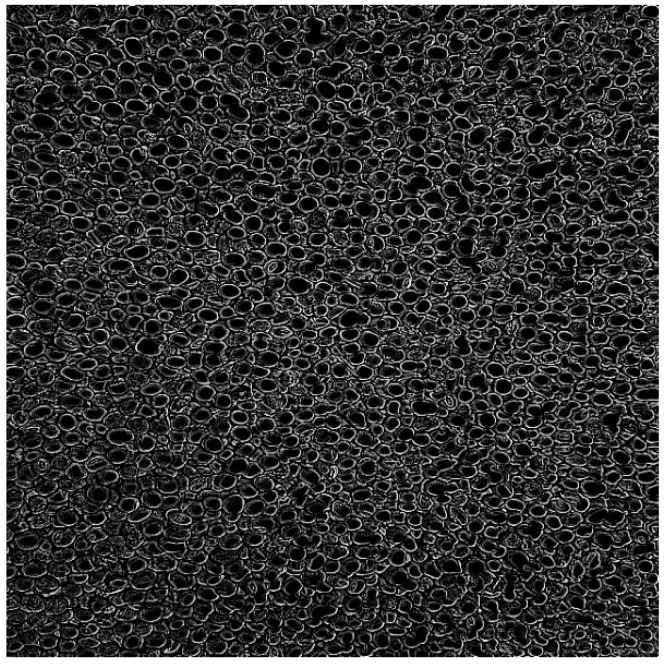
Sobel operator edge detection.

**Figure 5 sensors-25-04958-f005:**
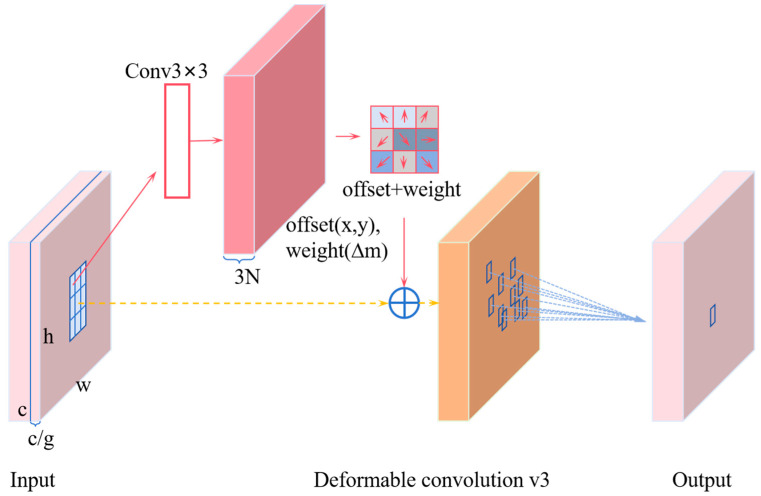
DCNv3.

**Figure 6 sensors-25-04958-f006:**
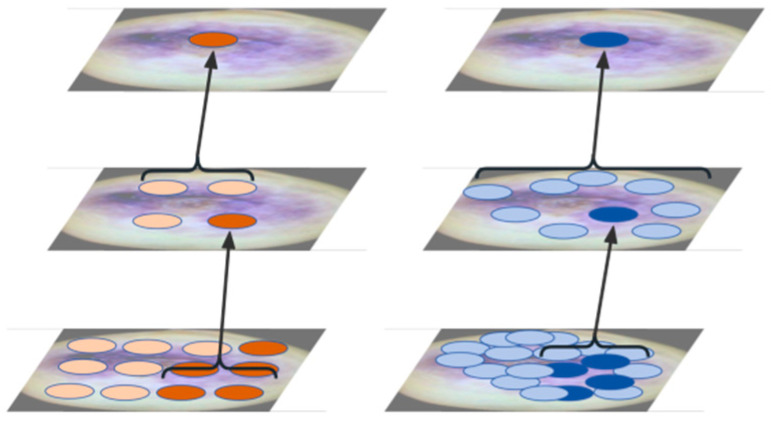
Deformable Convolutional Kernel Convolution ((**left**) Conv, (**right**) DConv).

**Figure 7 sensors-25-04958-f007:**
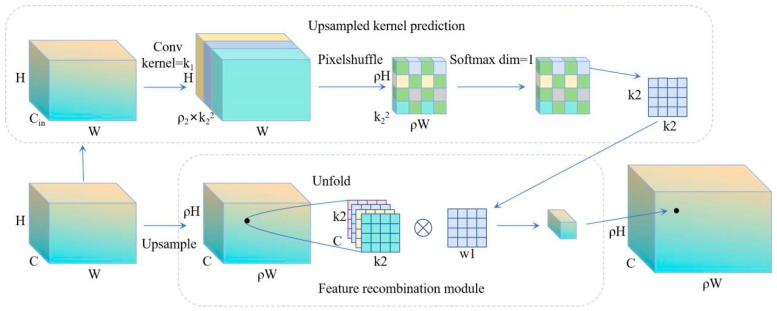
CARAFE Principle of operation.

**Figure 8 sensors-25-04958-f008:**
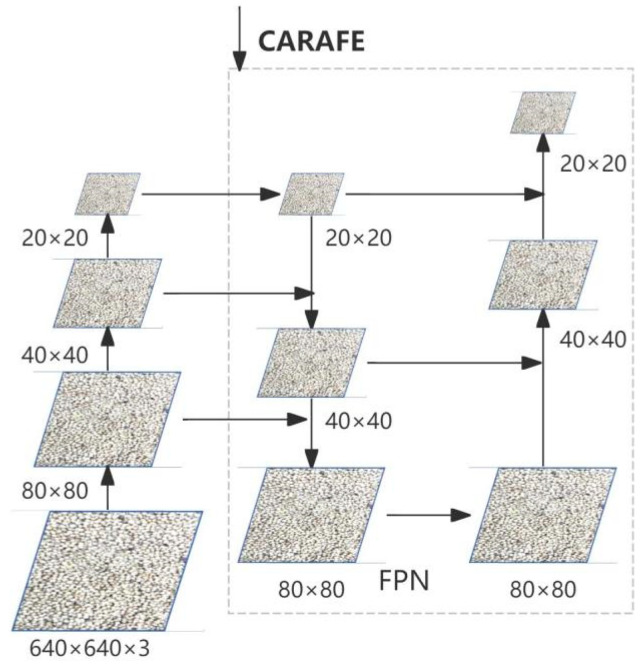
Overall structure of the CARAFE upsampling network.

**Figure 9 sensors-25-04958-f009:**
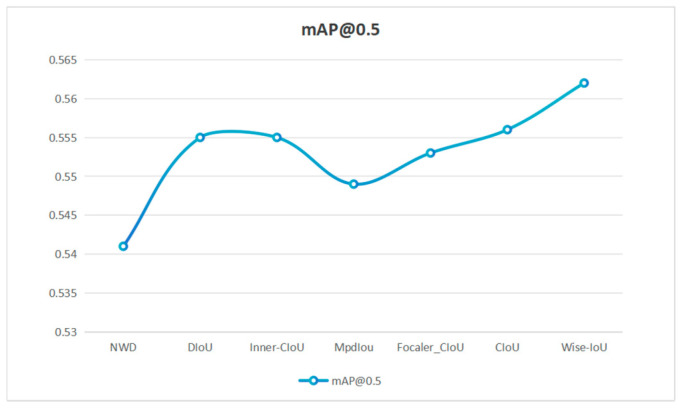
Effect of different loss functions.

**Figure 10 sensors-25-04958-f010:**
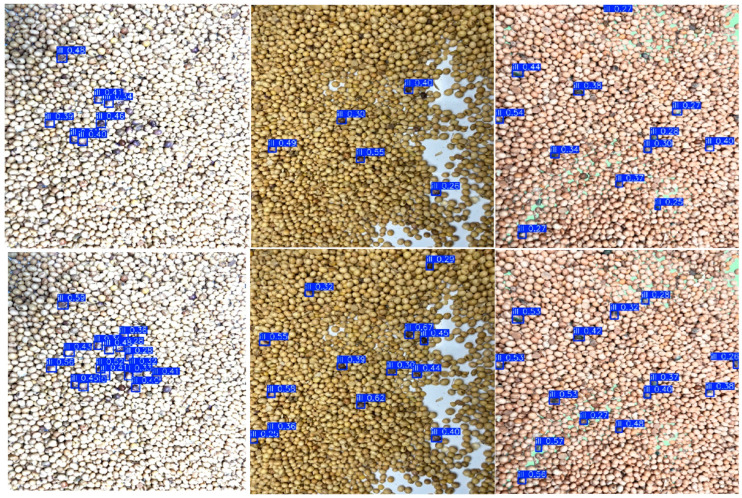
Comparison Effect YOLOv8n (**top**) and YOLOv8-ECCI (**bottom**).

**Figure 11 sensors-25-04958-f011:**
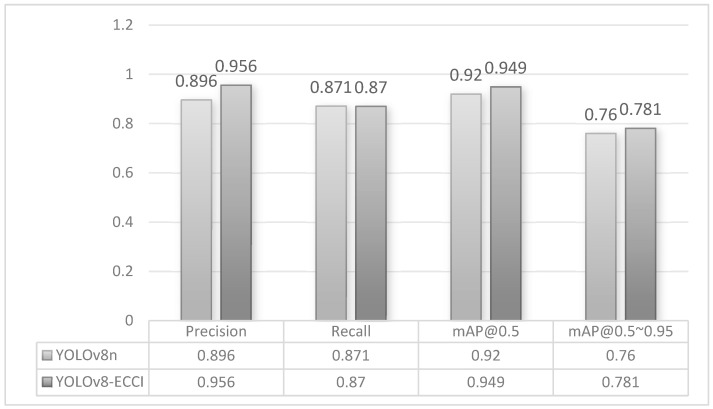
Model comparison results.

**Figure 12 sensors-25-04958-f012:**
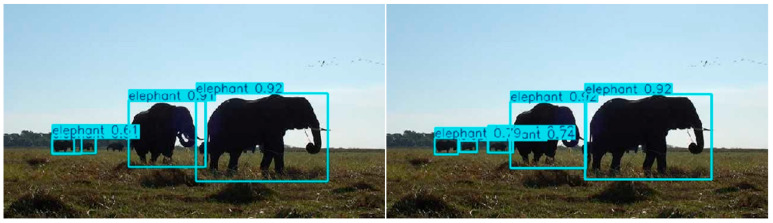
Comparative effects of generalized model tests YOLOv8n (**left**) and YOLOv8-ECCI (**right**).

**Table 1 sensors-25-04958-t001:** Sobel operator template.

Gx			Gy		
−1	0	1	−1	−2	−1
−2	0	2	0	0	0
−1	0	1	1	2	1

**Table 2 sensors-25-04958-t002:** Training parameter settings.

Parameter	Value	Parameter	Value
epochs	300	optimizer	SGD
imgsz	640	lrf	0.01
batch	32	box	0.75
workers	4	momentum	0.937
patience	100	weight_decay	0.0005
cls	0.5	warmup_momentum	0.8

**Table 3 sensors-25-04958-t003:** Comparative test results.

Models	Precision	Recall	mAP@0.5	mAP@0.5~0.95	Parameters/MB	GFLPOs
YOLOv5s	0.647	0.518	0.538	0.186	4.5	5.8
YOLOv9	0.641	0.510	0.536	0.196	12.7	22.1
YOLOv8n	0.629	0.528	0.540	0.190	5.4	6.8
YOLOv10	0.618	0.517	0.522	0.188	5.5	8.2
YOLOv11	0.646	0.527	0.560	0.198	5.2	6.3
YOLOv8-ECCI	0.718	0.576	0.627	0.267	6.0	8.2

**Table 4 sensors-25-04958-t004:** Ablation test.

Models	Precision	Recall	mAP@0.5	mAP@0.5~0.95	Parameters	GFLPOs
YOLOv8n	0.629	0.528	0.540	0.190	5.4	6.8
YOLOv8n + EIEStem	0.653	0.523	0.546	0.195	6.0	8.3
YOLOv8n + C2f_DCNv3	0.666	0.520	0.555	0.200	5.8	8.0
YOLOv8n + CARAFE	0.669	0.527	0.552	0.199	6.3	8.4
YOLOv8n + EIEStem + DCNv3	0.663	0.519	0.542	0.199	5.7	8.0
YOLOv8n + EIEStem + CARAFE	0.648	0.528	0.551	0.199	6.3	8.6
YOLOv8n + DCNv3 + CARAFE	0.636	0.556	0.559	0.200	6.0	8.0
YOLOv8n + EIEStem + DCNv3 + CARAFE	0.684	0.556	0.556	0.201	6.0	8.2
YOLOv8-ECCI	0.718	0.574	0.627	0.267	6.0	8.2

## Data Availability

The data presented in this study are openly available in Zenodo at https://doi.org/10.5281/zenodo.17232313.

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
