# Peer review of "Enhanced YOLOv8-ECCI Algorithm for High-Precision Detection of Purple Spot Disease in Soybeans"

_sensors, 2025, doi:10.3390/s25164958_

Round 1

Reviewer 1 Report

Comments and Suggestions for Authors

The relevance of this topic is clear: disease detection in seeds is challenging, particularly when sample sizes are small. Therefore, developing new methods is of significant interest.

However, this raises questions about the validity of the conclusions.

  1. How statistically significant and non-random are the reported improvements of a few percentage points? Why wasn't k-fold validation considered, given the small size of the dataset?
  2. Additional clarification is needed regarding the parameter choices in Table 2. How were these parameters selected, and how robust are the results with regard to these choices?
  3. There are significant issues with ablation study in Table 4. The impact of different improvements appears inconsistent. It is puzzling that a single improvement may have a positive effect on the results, yet when combined with another improvement, performance worsens.
  4. Several questions arise regarding the choice of baseline method. For example, why wasn't YOLOv11n, which already exists, included in the comparison?
  5. As far as I understand, lightweight models were selected for comparison, yet the computational costs are not discussed. How do the proposed modifications affect this aspect, considering not just the number of parameters, but also computational time?
  6. Is a precision below 70% sufficient for disease detection? If not, more heavy models should be considered to understand the potential.

Finally, the choice of the second dataset related to animals seems questionable, moreover, only the results are presented without any description of the example. Which specific properties of the datasets were the proposed improvements aimed at? Are they similar in any way? Why wasn’t another seed database considered?

Reviewer 2 Report

Comments and Suggestions for Authors

This paper proposes an improved YOLOv8-ECCI algorithm for seed-level high-precision detection of soybean purple spot disease. The research addresses core challenges such as small sample overfitting, spectral interference, and dense occlusion. By introducing modules like EIEStem, C2f-DCNv3, CARAFE, and Wise-IoU, the detection performance has been significantly enhanced. The paper has great application value and research significance. However, before it is accepted, I think some problems still need to be solved. Here are my comments:
In Section 3.4, when describing the CARAFE module, it does not fully explain how it dynamically generates upsampling kercores to handle soybean features. We only see the author's discussion on the perceptual characteristics of this module
Table 3 only compares the YOLO series models. We believe that the authors need to add horizontal model comparisons
3. It is suggested that the author optimize the abstract discussion and supplement the descriptions of parameters, reasoning speed and other indicators
4. Please add more research background on the seed level and literature related to seed-related diseases in the introduction section

Round 2

Reviewer 1 Report

Comments and Suggestions for Authors

The authors answered most of my questions; thank you. However, one important question remains unanswered.

Let me return to to the question about the ablation test. It needs clarification and explanation of Table 4.
The authors write:
"Table 4 shows that integrating the EIEStem module into the Backbone consistently improves performance metrics. "
From Table 4, precision (the first number) is better, but recall (the second number) is not:
YOLOv8n 0.661 0.509 0.547 0.196 11.47
YOLOv8n+EIEStem 0.687 0.504 0.553 0.203 11.67
The authors write:
"In contrast, applying the C2f-DCNv3 module alone does not produce a significant performance gain."
From Table 4, recall is better, but precition is not:
Why do the authors claim the opposite influence of EIEStem  and C2f-DCNv3?

Then the authors write:
"The experimental results further indicate that CARAFE offers even greater benefits when combined with the EIEStem and C2f-DCNv3 modules, and when paired with the Wise-IoU loss function."
YOLOv8n 0.661 0.509 0.547 0.196 11.47 
YOLOv8n+CARAFE 0.655 0.535 0.556 0.205 12
YOLOv8n+EIEStem+CARAFE 0.657 0.53 0.551 0.198 12

YOLOv8n+DCNv3+CARAFE 0.677 0.512 0.545 0.198 21.9
YOLOv8n+EIEStem+DCNv3+CARAFE 0.684 0.526 0.556 0.201 11.05
Recall for YOLOv8n+CARAFE is  0.535, while recall for YOLOv8n+EIEStem+DCNv3+CARAFE is 0.528.
Thus, please clarify and indicate which metric you consider as performance.
